# Genetic Background of Macular Telangiectasia Type 2

**DOI:** 10.3390/ijms26020684

**Published:** 2025-01-15

**Authors:** Ajda Kunčič, Mojca Urbančič, Darja Dobovšek Divjak, Petra Hudler, Nataša Debeljak

**Affiliations:** 1Medical Centre for Molecular Biology, Institute of Biochemistry and Molecular Genetics, Faculty of Medicine, University of Ljubljana, Vrazov trg 2, 1000 Ljubljana, Slovenia; ajda.kuncic@mf.uni-lj.si (A.K.); petra.hudler@mf.uni-lj.si (P.H.); 2Faculty of Medicine, University of Ljubljana, Vrazov trg 2, 1000 Ljubljana, Slovenia; mojca.urbancic@kclj.si (M.U.); darja.dobovsekdivjak@kclj.si (D.D.D.); 3Eye Hospital, University Medical Centre Ljubljana, Grablovičeva 46, 1000 Ljubljana, Slovenia

**Keywords:** retina, macular disorder, macular telangiectasia type 2, MacTel, complex disease, genetic predisposition, genomic biomarkers

## Abstract

Macular telangiectasia type 2 (MacTel) is a slowly progressive macular disorder that is often diagnosed late due to the gradual onset of vision loss. Recent advances in diagnostic techniques have facilitated earlier detection and have shown that MacTel is more common than initially thought. The disease is genetically complex, and multiple variants contribute incrementally to the overall risk. The familial occurrence of the disease prompted the investigation of the genetic background of MacTel. To better understand the molecular milieu of the disease, a literature review of the clinical reports and publications investigating the genetic factors of MacTel was performed. To date, disease-associated variants have been found in genes involved in amino acid (glycine/serine) metabolism and transport, urea cycle, lipid metabolism, and retinal vasculature and thickness. Variants in genes implicated in sphingolipid metabolism and fatty acid/steroid/retinol metabolism have been found in patients with neurological disorders who also have MacTel. Retinal metabolism involves complex biochemical processes that are essential for maintaining the high energy requirements of the retina. Genetic alterations can disrupt key metabolic pathways, leading to retinal cell degradation and the subsequent vision loss that characterizes several retinal disorders, including MacTel. This review article summarizes genetic findings that may allow MacTel to be further investigated as an inherited retinal disorder.

## 1. Introduction

Macular telangiectasia type 2 (MacTel) is a bilateral retinal disease with a late onset. Symptoms typically appear over the age of 40 years. The disease affects people all over the world, with no racial or gender predilection [1,2]. It was originally thought to be a disease of the retinal vessels. However, with the development of diagnostic imaging techniques, it has been shown to be primarily the consequence of neurodegenerative changes in the retina, with vascular changes appearing later in disease development. Due to the low level of awareness of the disease, many patients with earlier stages of the disease remain undiagnosed. For the same reason, some patients with later stages of the disease are misdiagnosed with other macular disorders.

The diagnosis of MacTel is based on clinical examination and multimodal imaging diagnostics, e.g., color fundus photography, fundus autofluorescence, fluorescein angiography, optical coherence tomography (OCT), and OCT-angiography [3]. The heterogeneity of the clinical presentation often makes diagnosis difficult and challenging. Typical clinical findings include reduced retinal transparency, crystalline deposits, mildly ectatic capillaries, blunted venules, photoreceptor loss, retinal pigment plaques, intraretinal cavities, lamellar or full-thickness macular holes, and neovascular complexes in the macula [3]. Observational studies have increased our knowledge of this disease. Vision loss primarily affects patients’ ability to read, but more extensive vision loss may occur as the disease progresses. Structural changes have been studied in detail and linked to function. Furthermore, certain systemic conditions, such as diabetes mellitus type 2 (T2DM), hypertension, heart disease, and thyroid disease, have been found to be more common in patients with MacTel [1,4,5,6,7,8,9,10]. Neurodegenerative changes are thought to be the primary manifestation affecting vision, with concomitant vascular aberrations that may lead to neovascularization as the disease progresses. Secondary vascular involvement in some cases may be related to the increased occurrence of T2DM, hypertension, and coronary artery disease in patients [11]. MacTel also co-occurs with neurological disorders, indicating a complex interplay of genetic variants and the involvement of different genes, leading to different neurological manifestations and phenotypes. For example, comorbidity with hereditary sensory and autonomic neuropathy type 1 (HSAN1) [12,13,14,15] and autosomal recessive spastic paraplegia type 56 (SPG56) [16] has been reported. The classification of the disease has evolved with the development of diagnostic techniques (Table 1). 

Originally, idiopathic juxtafoveolar retinal telangiectasia (IJRT) were classified into three groups: group 1, group 2 (with sub-groups A and B), and group 3. Sub-group 2A (MacTel) was further sub-divided into five stages [2]. Yannuzzi and colleagues revised this classification. Group 1 was named type 1 or aneurysmal, while group 2A was named type 2 or perifoveal and divided into non-proliferative and proliferative perifoveal telangiectasia. Groups 2B and 3 were omitted. [17]. The MacTel Research Group has recently developed an up-to-date classification of MacTel disease with 7 grades by incorporating image analysis results from novel imaging modalities. A revised classification that incorporates newer imaging modalities better describes the structural changes and is likely to facilitate a better description of the grades and progression of MacTel, as well as communication between clinical and basic science researchers [18]. While intravitreal treatment with vascular endothelial growth factor inhibitors is available to control exudative neovascularization [19], there are currently no therapies for the neurodegenerative changes associated with the disease.

The prevalence of MacTel is estimated at 0.0045–0.1% (4.5–100 cases per 100,000 individuals) [20,21]. Advances in diagnostic approaches have made it possible to recognize milder, earlier forms of the disease that were overlooked in the past, hence the wide range of the prevalence of this disease. It is now generally accepted that MacTel is more common than originally thought. The apparent genetic penetrance of MacTel is of great importance for gene discovery studies and for clinical risk assessment of family members of affected individuals. A penetrance of 0.35 was determined by sibling analysis and 0.55 by parent analysis. The combination of sibling and parent analyses resulted in an apparent penetrance of 0.38 [22]. The occurrence of the disease in multiple family members has prompted several investigations into the genetic background of MacTel. The initial studies analyzing candidate genes and potential disease-associated loci did not identify disease-associated genetic factors, indicating that a more complex genetic background is responsible for the disease mechanism [23,24,25,26]. Further studies utilizing genome-wide association studies (GWAS) have uncovered several genetic variants in a large cohort of patients [27,28]. The approaches used for the discovery of MacTel risk variants/candidate genes are revised in Figure 1. Although not characterized by any systemic pathology, the disease-associated metabolic profile has been described, outlining reduced serum serine and glycine levels and altered lipid metabolism in MacTel patients [12,29]. These findings are consistent with genetic studies that have identified variants in genes implicated in the glycine/serine metabolic pathway and lipid metabolism [29].

This article summarizes the genetic alterations associated with MacTel that have been reported up to 1 August 2024. The online databases PubMed and Google Scholar were searched for the relevant literature using the following keywords: “macular telangiectasia”, “macular telangiectasia type 2”, “MacTel”, plus “genetics”, “genomics” and “family members”. The titles and abstracts were screened first, followed by an eligibility assessment of the full text. Inclusion criteria: publications written in English referring to human studies were included. Exclusion criteria: Macular telangiectasia type 1 and 3, conference abstracts, and master’s and doctoral theses were not included in the review. The disease associated-variants are listed in the Appendix A and were visualized on the circular karyogram of the human genome using the “circlize” package in R (version 4.4.2) (Figure 2).

The gene names are consistent with the HUGO Gene Nomenclature Committee (HGNC) [32] and National Center for Biotechnology Information (NCBI) [33] databases. The variants are represented with rs numbers and include information about their position at a coding sequence, genome, and/or protein level (GRCh38.p14). Allele frequencies were collected from gnomAD for the total and European (Non-Finnish) ancestry group [34]. Variant type, consequence, and clinical significance were obtained from the NCBI databases ClinVar, dbSNP, and Ensembl Release 112 (May 2024) [35].

## 2. Key Genes, Genomic Regions, and Metabolic Pathways Associated with MacTel

### 2.1. Familial Occurrence of MacTel

Based on clinical reports of the disease in monozygotic twins and family members (Table 2), the involvement of genetic factors in the pathogenesis of MacTel was proposed.

In the late 1970s, the disease was first characterized as a distinct clinical entity. Hutton et al. (1978) examined four patients, including two sisters (aged 46 and 56 years), with an unusual form of retinal telangiectasia [36]. Chew et al. (1986) described the condition in the non-diabetic brother of a diabetic patient while reporting on five patients with parafoveal telangiectasia and mild diabetic retinopathy [6]. In an updated classification and follow-up study of IJRT, 3 of the 92 patients were siblings, and the disease was present in 2 of 89 families [2]. Isaacs and McAllister (1996) studied two sisters with bilateral IJRT over an 8-year period [37]. Oh and Park (1999) described vertical transmission in the 29-year-old daughter of a 58-year-old affected father [38]. At that time, the case was unique in that the disease was vertically transmitted and the fibrovascular tissue developed in the young proband. The authors assumed that inherited factors and T2DM predisposed the proband to develop early and severe fibrovascular tissue. Another report on the vertical transmission of MacTel and T2DM in three families was published in 2012 [39]. Vertical transmission in families suggests a dominant inheritance [3]. In 2000, bilateral juxtafoveolar telangiectasia was reported in monozygotic twins (64-year-old sisters), raising the question of genetic influences in the pathogenesis of the disease [40]. Later reports of affected monozygotic twins were published in 2005 (68-year-old sisters) [41], 2007 (63-year-old sisters) [42], and 2009 (56-year-old sisters and 56-year-old brothers) [11], strengthening the theory of the genetic component of the disease.

The discovery of affected asymptomatic family members indicates that the condition is more widespread than previously thought. The identification of asymptomatic family members or members with early-stage disease is currently limited by the lack of non-invasive diagnostic techniques that can detect subtle changes characteristic of early-stage disease. In addition, studies support the hypothesis that patients with MacTel have a genetic predisposition to the disease [11,24]. However, it has been suggested that, in monozygotic twins, a secondary factor contributes to the development of the clinically evident features. The influence of environmental factors in combination with genetic factors could potentially explain the phenotypic discordance between monozygotic twins with differences in disease stage. The local intraocular environment might also be important. Particular emphasis should be given to epigenetic variations that could establish a pathogenic link between external and internal factors.

### 2.2. Candidate-Gene Screening Analysis

Barbazetto et al. (2008) investigated variants in ATM serine/threonine kinase (*ATM*), age-related macular degeneration (AMD)-associated variants in complement factor H (*CFH*), complement factor B (*CFB*), and 10q26 loci in a cohort of 30 MacTel cases [23]. The *ATM* gene is crucial for DNA damage response and repair (genome stability), while *CFH* and *CFB* are involved in the alternative pathway of complement activation and play a role in the immune response [43,44]. Variations in *ATM* were screened using a combination of denaturing high-performance liquid chromatography and direct sequencing. Major AMD-associated alleles in *CFH*, *CFB,* and 10q26 were screened using polymerase chain reaction restriction fragment-length polymorphism. Screening of *ATM* revealed amino acid changes in 23 patients of European–American descent. Screening revealed a null allele rs587780612 (*ATM*) in 2 of 23 (8.7%) patients of European descent; previously disease-associated missense alleles rs4986761 and rs1800057 (both in *ATM*) in 4 of 23 (17.4%) patients; and the common missense alleles rs4986761, rs1800057, rs1801516, rs148993589, rs1800058, rs2234997, and rs3218695 (all in *ATM*) in 7 of 23 (30.4%) patients (Figure 3, Appendix A). Interestingly, no variants in the *ATM* were identified in patients of Asian or Hispanic descent. The frequencies of the major AMD-associated alleles in the *CFH*, *CFB*, and 10q26 loci in the MacTel cohort were like those in the ethnically matched general population. Between 26 and 57% of the MacTel patients of European–American ancestry included in the study carried *ATM* alleles that could be disease-associated. Therefore, certain *ATM* variations may predispose people to the development of MacTel [23].

Parmalee et al. (2010) searched for the gene(s) responsible for MacTel by using a candidate-gene screening approach [25]. The candidate genes were selected based on the following criteria: (1) genes known to cause or be associated with diseases with phenotypes similar to MacTel, (2) genes with known function in the retinal vasculature or macular pigment transport, (3) genes that emerged from expression microarray data from mouse models designed to mimic MacTel phenotype characteristics, and (4) genes expressed in the retina that are also related to T2DM or hypertension, which have an increased prevalence in MacTel patients. Probands (8) from eight families with at least two affected individuals were screened by direct sequencing of 27 candidate genes, including the following: *AGGF1*, *ANG1*, *DKK1*, *FZD4*, *HIF1A*, *LRP5*, *NDP*, *PEDF*, *THBS1*, *TIE2*, *VHL*, *AGTRL1*, *APLN*, *CFB*, *LRG1*, *PLVAP*, *GTSP1*, *SCARB1*, *CCM2*, *IGFBP3*, *SSPN*, *TGFB2*, *AGTR1*, *ALDH3A2*, *OXGR1*, *SUCNR1*, and *VLDLR*. A total of 23 non-synonymous missense variants, 22 synonymous variants, and 61 intronic variants were identified. Of the non-synonymous variants, 12 missense variants were analyzed for co-segregation with the disease and/or allele frequency analysis using TaqMan assays in 400 MacTel cases and 368 controls. None of these variants were segregated with the disease. Only one variant, rs1695 (*GSTP1*), showed a trend toward a significant frequency difference between MacTel cases and controls, suggesting that it is a possible modifier but not a causative gene for MacTel (Figure 3, Appendix A). Furthermore, variant rs61735304 (*FZD4*), previously suggested to be a disease-causing variant in familial exudative vitreoretinopathy, was determined to be a rare, benign polymorphism [25].

Szental et al. (2010) conducted a case–control study of 39 MacTel cases and 21 controls to determine whether two variants of the *GSTP1* gene are associated with MacTel [26]. The Pi isoform of glutathione S-transferase (GSTP1) is a xanthophyll-binding protein (XBP). The uptake of xanthophyll pigment into the macula is thought to be facilitated by XBP, and patients with MacTel have decreased macular pigment centrally. Two typical functional polymorphic sites, rs1695 and rs1138272, in *GSTP1* were analyzed. The study found no statistically significant association between the variants and MacTel. However, a trend towards a greater frequency of the GG genotype of rs1695 (*GSTP1*) in cases (3/39; 8%) could be significant (Figure 3, Appendix A). The biological plausibility of impaired macular pigment uptake in MacTel makes *GSTP1* an excellent candidate gene for further investigation in larger cohorts of patients and unaffected controls [26].

GSTP1 plays a crucial role in the detoxification of electrophilic compounds, including carcinogens, therapeutic drugs, environmental toxins, and products of oxidative stress, through conjugation with glutathione [45]. The rs1695 (c.313A > G) results in an amino acid substitution at position 105 of the GTSP1 protein (p.Ile105Val). This single nucleotide polymorphism (SNP) affects enzyme activity and specificity, which may have an impact on detoxification efficiency [46,47]. Due to the high metabolic demands and oxidative stress in the retina [48], *GSTP1* variants could be further investigated for their influence on retinal health.

### 2.3. Genome-Wide Linkage Analysis

The only one genome-wide linkage analysis did not reveal any disease variants (Figure 4) [24].

As part of the MacTel project, 17 pedigrees with multiple affected family members were identified from a large cohort of MacTel patients, consistent with autosomal dominant segregation with reduced penetrance [24]. A genome-wide linkage analysis of seventeen families with a total of 71 individuals (including 45 affected (MacTel cases) or possibly affected individuals) identified a single peak with a multi-point LOD score of 3.45 on chromosome 1 at 1q41–42 under a dominant model. Recombination mapping defined a minimal candidate region of 15.6 Mb (214.32–229.92 Mb), encompassing the 1q41–42 linkage peak. Interestingly, Sanger sequencing of the top 14 positional candidate genes, including *DISP1*, *TLR5*, *SUSD4*, *CAPN8*, *CAPN2*, *TP53BP2*, *FBXO28*, *DEGS1*, *NVL*, *CNIH4*, *WDR26*, and *CNIH3* and the micro RNA *MIR320B2* under the linkage peak, revealed no causal variants in these pedigrees [24].

However, according to the UCSC Genome Browser, this region contains more than 1700 transcripts (including gene isoforms and non-coding RNA transcripts) [49,50]. Therefore, the disease could be driven by variants in and/or aberrant expression of either of these coding and non-coding genes. Furthermore, it would be interesting to identify the minimal driver or initiation aberrations that, together with passenger alterations, contribute to the different clinical courses of the disease.

### 2.4. Genome-Wide Association Studies

The first GWAS identified three genome-wide (GW) significant (*p* < 5 × 10^−8^) associations, i.e., rs477992 (*PHGDH*), rs715 (*CPS1*), and rs73171800 (ENSG00000271904; *TMEM161B–MEF2C*) [28]. The discovery stage included genotype data for 6,312,048 SNPs in 476 MacTel cases and 1733 controls of European descent. Variants at six loci surpassed the GW significance threshold. Three of these were discounted (1p36.22, 3p24.1, and 7p21.3) after failing the technical validation step. The remaining three loci (1p12, 2q34, and 5q14.3) included 149 SNPs that reached GW significance. Seven of the 149 SNPs, including rs539708, rs666930, rs483180 (all in *PHGDH*); rs715, rs4673553 (both in *CPS1*); rs17478824 (ENSG00000271904; *TMEM161B–MEF2C*), and rs73173548 (*MIR9-2HG*) were genotyped with TaqMan assays in the replication stage at an independent cohort of 172 MacTel cases and 1134 controls (European descent) [28].

At locus 5q14.3, the strongest associated variant in the discovery stage was rs73171800 (ENSG00000271904; *TMEM161B–MEF2C*) among 116 SNPs in this region. The rs73171800 is in strong linkage disequilibrium (LD) with rs17478824 (ENSG00000271904; *TMEM161B–MEF2C*), rs73173548 (*MIR9-2HG*), rs2194025 (*TMEM161B*–*MEF2C*), and rs17421627 (*MIR9-2HG*) (all among the GW significant SNPs at this locus). For further analysis in the replication stage in an independent cohort of MacTel patients, the variants rs17478824 (ENSG00000271904; *TMEM161B–MEF2C*) and rs73173548 (*MIR9-2HG*) were selected. The analyses in the independent cohort strongly confirmed that minor alleles of these two variants increase the risk of developing MacTel [28].

At locus 2q34, the strongest associated variant was rs715 (*CPS1*) out of 21 SNPs discovered in this region. Together with another significant variant, rs4673553 (*CPS1*), these two were selected for replication in the independent cohort. The major allele of rs715 (*CPS1*) confers risk for MacTel. Females were at nearly two times greater risk of MacTel for each additional copy of the rs715 (*CPS1*) risk allele [28].

At locus 1p12, the most significant variant was rs477992 (*PHGDH*), followed by rs478093 (*PHGDH*) out of the 12 significant SNPs at this locus. The signal was replicated with the variant rs483180 (*PHGDH*) in the independent cohort. Minor alleles of rs477992 and rs478093 (both in *PHGDH*) confer risk for MacTel and have been associated with decreased serine levels [28].

A total of 25 loci showed suggestive evidence of association with MacTel (1 × 10^−5^ > *p* ≥ 5 × 10^−8^). These loci were explored for overlap with genetic regions previously associated with serine and glycine levels, and the loci at 3q21.3 and 7p11.2 were identified. On 3q21.3, rs9820286 and rs9880406 (both between *ALDH1L1*–*KLF15*) are suggestively associated with MacTel, and the association with MacTel for rs9880406 (*ALDH1L1*–*KLF15*) was replicated in the independent cohort. At 7p11.2, suggestive significance was found for variants rs4947534 and rs4948102 (both in *PSPH*) (the strongest signal at this locus). The variants selected for replication in the independent cohort were rs4535700 and rs11238389 (both in *PSPH*). The variants rs4947534, rs4948102, and rs11238389 (all in *PSPH*) are in perfect LD and in high LD with rs4535700 (*PSPH*). The minor allele of these variants is associated with MacTel risk, and, similarly, the minor allele of rs4947534 (*PSPH*) is associated with reduced serine levels and increased homocysteine levels [28].

The identified genetic loci accounted for only 5% of the estimated heritability. By increasing the sample size, more disease-associated loci were discovered. The second GWAS included 1067 MacTel cases and 3799 controls (of European descent, with a higher proportion of females) [27]. The extended study confirmed all three previously reported loci (1p12, 2q34, and 5q14.3) and identified seven novel GW-significant loci (2p14, 3p24.1, 3p21.31, 7p11.2, 9p22.3, 10q21.3, and 19p13.2) and seven novel suggestively significant loci (1q32.3, 2p23.2, 6q13, 8q24.21, 11p13, 17q25.1, and 19p13.2). After quality control and imputation, genotype data for 7,289,516 SNPs were included in GW association testing. The MacTel consortium samples were genotyped on the Illumina Human Omni5-Exome-4 Array or the Illumina Global Screening Array; the Aged-Related Eye Diseases Study controls were genotyped on the Illumina Omni2.5 BeadChip Array; and the Australian Twinning controls were genotyped on the Illumina Global Screening Array. Two GWAS were performed, one for the entire cohort (full-cohort GWAS) and one restricted to individuals of European descent (European cohort GWAS). In the full cohort analysis, eleven independent GW significant disease associations were identified in ten regions, of which seven results in six regions were preserved in the European cohort GWAS [27].

A study by Bonelli and colleagues further confirmed the involvement of specific metabolic pathways in MacTel pathogenesis (Figure 5). Variants were found in genes implicated in the glycine/serine metabolic pathway (*PHGDH* and *PSPH*), metabolite transport (*SLC1A4* and *SLC6A20*), the urea cycle (*CPS1*), retinal vasculature and thickness (*MIR9-2HG)*, and other genes such as *TTC39B*, *REEP3*, or intergenic regions between *SLC4A7*–*EOMES* and *CERS4*–*FBN3* [27].

The variants on 2q34 and 5q14.3 are in LD with the previously identified tagging SNPs rs715 (*CPS1*) and rs73171800 (ENSG00000271904; *TMEM161B–MEF2C*), while the most significant SNP rs146953046 (*PHGDH*) in the 1p12 region is in low LD with and independent of the previously identified SNP rs477992 (*PHGDH*). Conditional analysis of this locus revealed a second signal in which the most significant SNP, rs532303 (*PHGDH*), is in strong LD with the previously identified SNP rs477992 (*PHGDH*) [27].

Conditional regression analysis (GWAS conditioning on the metabolic polygenic risk scores) identified four GW significant peaks. Original signal rs73171800 (ENSG00000271904; *TMEM161B*–*MEF2C*) on locus 5q14.3 and three novel disease loci independent of endogenous serine biosynthetic capacity, including rs35356316 (*EOMES*–*SLC4A4*) on 3p24.1 and rs36259 (*CERS4*) and rs4804075 (*NFILZ*) on 19p13.2, were found. Genetically induced serine deficiency was suggested to be the primary causal metabolic driver of MacTel, with a contribution of genetic T2DM risk. In contrast, glycine, threonine, and retinal vascular traits were deemed unlikely to be causal for MacTel [30].

GWAS have significantly advanced our understanding of the genetic basis of many complex traits and diseases. However, they come with several limitations, including the following: (1) GWAS identify common genetic variants, which usually have a small effect on disease, while they are not effective in identifying rare variants with larger impact on the phenotype; (2) GWAS often identify loci associated with traits, but not the specific causal variants; and (3) complex traits and diseases are often influenced by multiple genetic and environmental factors, making it challenging for GWAS to capture the full genetic architecture. The MacTel-associated genomic variants discovered through GWAS are listed in Appendix A and shown on Figure 5.

### 2.5. Next-Generation Sequencing Studies

Genes and variants that lead to the reduced serine levels observed in MacTel patients remain elusive. Using whole-exome sequencing data from 793 MacTel cases and 17,610 controls, the disease-associated variants were identified. Collapsing analysis identified 22 rare variants in *PHGDH* in 29 MacTel cases (Figure 2, Appendix A). The resulting functional defects induced by the most common missense variant rs139063843 (*PHGDH*) were strongly associated with the decrease in serine biosynthesis and the accumulation of deoxysphingolipids (deoxySLs) in retinal pigment epithelium (RPE) cells [31].

### 2.6. Implications of Risk-Associated Metabolic Pathways on MacTel Pathogenesis

The pathogenetic mechanism leading to MacTel development is largely unknown. Most genetic studies clearly indicate that genetic differences in glycine/serine metabolism, the urea cycle, and metabolite transport could be the possible background for MacTel development. In addition, the differences in disease progression suggest that certain combinations of genetic aberrations in conjunction with lifestyle and other factors influence the trajectory of disease severity and progression over time.

In addition to genetic studies, causative disease mechanisms have also been investigated using other omics approaches (metabolomics and proteomics), and multiple models have been developed, including induced pluripotent stem cells, organoids, and animals (mouse and rat models) [51,52,53,54,55,56].

For example, a metabolomics study of 60 MacTel cases and 58 controls confirmed reduced serum serine and glycine levels in the patients, as well as altered lipid metabolism with elevated phosphatidylethanolamines, long-chain fatty acids, diacylglycerols, and monoacylglycerols [29]. They also found out that methionine, betaine, and the urea cycle intermediates were decreased. This finding prompted further investigation focusing on key metabolites of methionine and one-carbon metabolism, such as homocysteine, S-adenosylmethionine, S-adenosylhomocysteine, and total glutathione levels. Methylation-associated metabolite levels were investigated in 29 MacTel cases, revealing that homocysteine could be an important metabolite discerning MacTel patients from other diseases with specific metabolite-methylation patterns [57]. In another study utilizing cross-platform analyses of associations between genetic effects and the blood levels of 174 metabolites, the researchers found that higher serine levels reduced the likelihood of MacTel development by 95%, indicating that even slight changes in the concentrations of key metabolites can have profound effects on particular cell types [58].

Research at the proteome level revealed the aberrations in protein expression directly in the vitreous and confirmed the involvement of several metabolic pathways directly or indirectly involved in amino acid and lipid metabolism, including glycolytic anaplerotic reactions [59].

Overall, the studies indicated a strong link between MacTel, altered serine metabolism, and elevated levels of atypical deoxySLs. Some studies have already established a genetic predisposition to altered amino acid homeostasis and identified variants that could be responsible for reduced serine levels in patients [12,27,28,31]. The functional *PHGDH* variants identified in MacTel patients may result in decreased serine biosynthesis and the accumulation of deoxySLs [31].

Serine is considered a conditionally essential amino acid for some tissues that have a high demand for protein synthesis, organelle turnover, and lipid homeostasis. Phosphoglycerate dehydrogenase (PHGDH) catalyzes the first committed step of de novo serine biosynthesis from 3-phosphoglycerate [60]. Interestingly, the ability of cells to synthesize serine from glycolytic intermediates renders serine a non-essential amino acid; however, in recent years, its importance has been recognized by studies of malignant diseases (for a review, see [61,62]). Serine has been shown to be an important metabolite, intertwining several essential metabolic pathways, including glycolysis, the folate cycle, and one-carbon metabolism, as well as the metabolism of sphingolipids, phospholipids, and sulfur-containing amino acids [60,63]. Serine is involved in cysteine biosynthesis via the transsulfurylation pathway; moreover, cysteine in turn is essential to produce glutathione. Glutathione and glutathione-associated enzyme systems in the retina are one of the most important antioxidants that minimize the damage caused by photooxidation products [64].

The retina is very metabolically active, and the transport of serine across the blood–retinal barrier into the RPE or across the endothelial cells to the neurosensory retina is supposedly inadequate. The additional supply of serine to the photoreceptors and the inner retina is provided by the RPE and the Müller glial cells (MGCs). Both cells can synthesize serine. MGCs in the macula show increased PHGDH levels, glutathione and glycine production, and are more susceptible to induced stress [65]. Many retinopathies are associated with the loss of MGCs, including MacTel, which leads to impaired PHGDH activity and reduced serine levels. The supply of glycine and glutathione, and thus serine, is essential for the reactive oxygen species mitigation system of the rods and cones. Consequently, reduced serine levels have a negative impact on these repair mechanisms. Lower levels of D-serine, which functions as a neurotrophic factor and co-agonist for N-methyl-D-aspartate receptors, probably promote neurodegenerative changes in MacTel patients [65].

Phosphatidylserine is important for the recognition and phagocytosis of the correct portion of the outer segments. Inefficient or delayed phagocytosis of the photoreceptor outer segments can lead to abnormalities of the neurosensory retina and RPE. Severe serine deficiency and dysregulated lipid metabolism can lead to increased and leaky vasculature and contribute to vascular abnormalities [65].

Prolonged serine deficiency leads to the deregulation of amino acid homeostasis, which can result in the generation of non-canonical sphingolipids [66]. Glycine, alanine, cysteine, and threonine share structural similarities to alanine; in addition, in serine deficiency, serine palmitoyltransferase (SPT) can produce cytotoxic deoxySLs by using alanine or glycine as a substrate for condensation with palmitoyl-CoA [15,66,67,68]. DeoxySLs have been shown to be responsible for photoreceptor cell death in retinal organoids [12].

Studies also indicated that variants in metabolite transporters may be implicated in the pathological complexity of MacTel. Indeed, the *SLC1A4* gene (protein ASCT1) encodes the sodium-dependent transporter for neutral amino acids, which transports serine, alanine, cysteine, proline, hydroxyproline, and threonine, whereas XTRP3 (gene *SLC6A20*) mediates the sodium- and chloride-dependent uptake of amino acids such as proline, as well as glycine and N-methylated amino acids. Variations that impair the function of such transporters could further deregulate amino acid homeostasis in susceptible cells with specific requirements for certain amino acids, such as MGCs and RPE [12,31,69].

### 2.7. Association of MacTel with Comorbidities

MacTel co-occurs with some neurological and metabolic disorders. Therefore, genetic variants in one gene or interactions between variants in several genes together with environmental factors could contribute to the development of primary and associated diseases [3,11]. Such variants have been found in MacTel patients in subunits (SPTLC1 and SPTLC2) of the first enzyme in the ceramide/sphingolipid metabolic pathway (SPT) and in cytochrome P450 2U1 (*CYP2U1*), which is implicated in oxidation and fatty acid metabolism (Appendix A). SPT mutations are known to cause HSAN1, a rare condition that causes nerve damage in the extremities. The hallmark of the disease is loss of pain and temperature sensation in the distal parts of the lower limbs. The autonomic disturbances, if present, manifest themselves in the form of sweating abnormalities. A comparison of the enzymatic properties of eleven SPTLC1 and six SPTLC2 isoforms revealed distinct associations between the mutated proteins and the clinical phenotype, as well as the severity of the symptoms. Both isoforms are ubiquitously expressed and are involved in the condensation of palmitoyl-CoA and L-serine in the de novo synthesis of sphingolipids. Interestingly, most mutants showed increased activity and affinity for alanine and diminished specificity for serine, resulting in the overproduction of deoxySLs, detectable in the plasma of patients. Three mutant isoforms, p.Ser331Phe, p.Ser331Tyr (*SPTLC1*), and p.Ile504Phe (*SPTLC2*), produced both canonical C_18_-sphingolipids and anomalous C_20_-sphingolipids [70].

Nine of eleven patients with HSAN1, caused by SPT variants p.Cys133Tyr (*SPTLC1*) or p.Ser384Phe (*SPTLC2*), were found to have MacTel [12]. Additionally, the p.Ser384Phe variant (*SPTLC2*) has been reported in two generations of a single family [14]. An association between the p.Asn177Asp variant (*SPTLC2*) and MacTel was also found in a family member with multiple members of the family with HSAN1 [15]. Nevertheless, the link between HSAN1 and MacTel appears to be more complex and cannot be explained by the genetic variants alone, as patients with HASN1 (age 35–71 years) who have p.Cys133Tyr and p.Cys133Trp variants (both in *SPTLC1*), or p.Ala182Pro variant (*SPTLC2*) showed no clinical signs of MacTel [13]. A particular phenotype with MacTel was observed in the case of a neurologically asymptomatic boy with an unusual phenotype of *CYP2U1*-related macular dystrophy. Whole-exome sequencing identified a disease-causing pathogenic variant (p.Arg390Ter) in CYP2U1 causing SPG56 [16].

## 3. Conclusions

MacTel is a clinically heterogeneous and genetically complex macular disorder. The path to understanding the genetic complexity of the disease has not yet been paved. Translating this knowledge into clinically relevant information is a daunting task due to the complexity of the MacTel phenotype and the myriad of alterations that can affect gene expression and function. Based on the current state of research (summarized in Figure 6), it appears that the combined aberrations in serine metabolic pathways, as well as sphingolipid, and possibly urea and cholesterol metabolism may contribute to the initiation and development of the disease. 

Studies have shown that the most important genetic changes driving the course of the disease are most likely different variants in genes. It could also be that the disease is not caused by rare genetic variants, but by the particular combination of common genetic variants in certain genes of key metabolic pathways associated with MacTel pathogenesis. In addition to the variants in protein-coding genes, the variants in non-coding RNA molecules involved in the fine-tuning of the gene expression could also influence the delicate metabolite balance in the cells. Epigenetic changes such as DNA methylation and histone modifications, which can also aberrantly accumulate in response to oxidative stress, environmental stress, and other factors, might further contribute to neurodegeneration and abnormalities of MacTel. Taken together, these genetic and epigenetic factors, as well as individual lifestyle factors, including quality of nutrition, could account for differences in symptoms, time of onset, and disease severity. An ongoing phase 2a clinical study (NCT04907084) in MacTel patients is investigating the effect of serine supplementation and fenofibrate treatment; however, to our knowledge, no results have yet been published.

In this review, we focus on the current knowledge about the genetic background of the disease. Several studies address the metabolomics alternations in affected patients; however, they do not address the genetic background of these patients. It should be noted that systemic serine and glycine deficiency has already been strongly associated with metabolic syndrome and various macular and peripheral nerve disorders [12,71]. Emerging evidence clearly indicates a link between diabetic polyneuropathy and serine deficiency in diabetic mouse models, possibly due to altered glucose metabolism leading to altered levels of 3-phosphoglycerate, the main serine precursor [71,72]. Interestingly, low serine levels and higher toxic deoxySLs have been associated with HSAN1, and patients with HSAN1 often present with MacTel; however, patients with a primary diagnosis of MacTel rarely have HSAN1. It is not yet clear whether peripheral neuropathy may also be present in the primary MacTel background. Therefore, it appears that an imbalance of metabolites in specific cell types could underlie the disease and cause different symptoms in lieu of individual genetic and epigenetic frameworks. In the future, it would also be interesting to explore the specific metabolic requirements of particular cell types, as research indicates that some cells are particularly sensitive to metabolite fluctuations. The apparent functional dependence of supportive cells to retinal neurons on serine homeostasis may suggest that L-serine supplementation could be used to slow and possibly partially ameliorate vision loss in MacTel patients, as well as patients with other complex diseases at risk for serine deficiency. In addition, serine supplementation could reduce the oxidative stress and neurotoxicity of aberrant metabolic by-products and improve the antioxidant capacity of the retina.

Studies linking the metabolomic alternations to the genetic background of patients will help to further clarify the genetic background and underlying mechanisms of MacTel. This will enable better management of patients, early diagnosis, and the development of potential personalized treatments in the future.

## Figures and Tables

**Figure 1 ijms-26-00684-f001:**
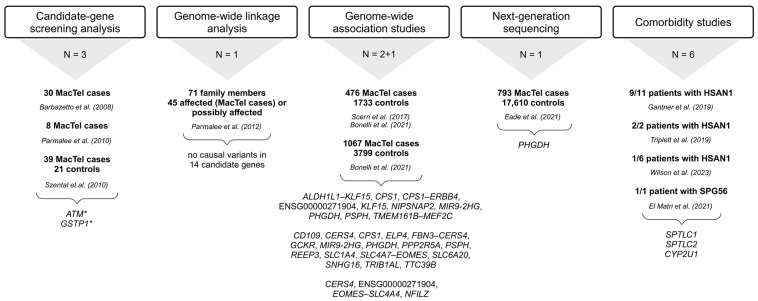
Summary of approaches used to identify MacTel-associated variants and genes [12,13,14,15,16,23,24,25,26,27,28,30,31]. Below each approach is the number of studies (N) utilizing it, followed by the number of study participants (MacTel cases and controls) (text in bold), the publication, and the list of genes with disease-associated variants. * *ATM* sequence variants could be disease-associated [23]; * *GSTP1* is a possible modifier but not a causal gene for MacTel [25]; trend towards greater frequency of GG genotype in cases [26]. The full names of the genes are provided in the Appendix A. Created in BioRender.

**Figure 2 ijms-26-00684-f002:**
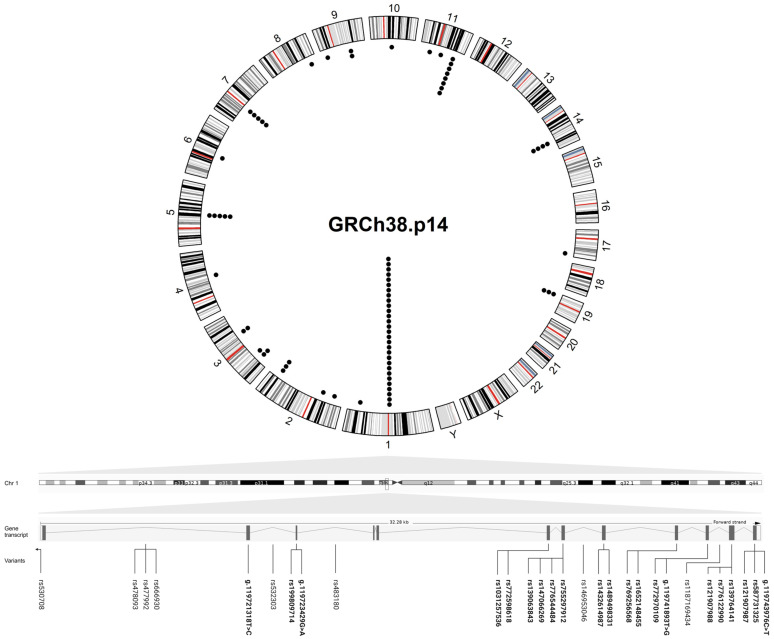
Circular plot showing the genomic location of the variants found in MacTel patients. The outer circle represents a karyogram of the human genome (GRCh38.p14) (red lines represent centromeres), while the inner circle shows the variants (each black dot corresponds to a variant). A linear transcript ENST00000641023.2 of *PHGDH* containing the highest number of disease variants is shown below the circular plot, with the intron and exon (rsIDs in bold) variants found in MacTel patients distributed to their position across the gene (the lines indicate the position of the variants in the gene transcript). The circular plot was visualized in R (version 4.4.2) and the figure was created in BioRender.

**Figure 3 ijms-26-00684-f003:**
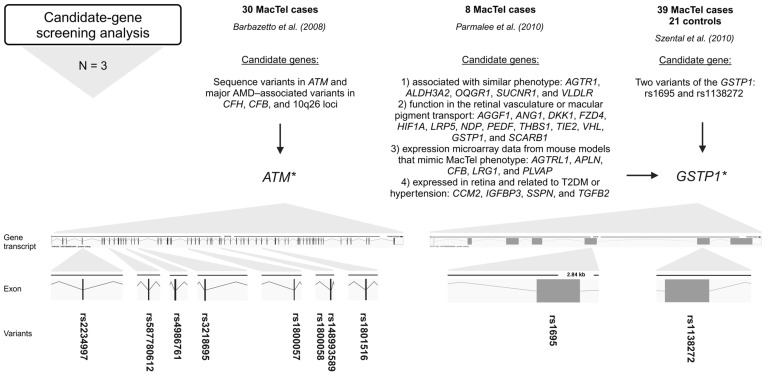
Variants (rsIDs in bold) identified in MacTel patients using candidate-gene screening analysis. AMD-associated variants in *ATM* and variants in *GSTP1* were found in MacTel patients [23,25,26]. Below the approach, the number of studies (N) is given, followed by the number of study participants (MacTel cases and controls) (text in bold), the publication, the candidate genes analyzed, and the variants found. * *ATM* sequence variants could be disease-associated [23]; * *GSTP1* is a possible modifier but not a causal gene for MacTel [25]; trend towards greater frequency of GG genotype in cases [26]. Created in BioRender.

**Figure 4 ijms-26-00684-f004:**
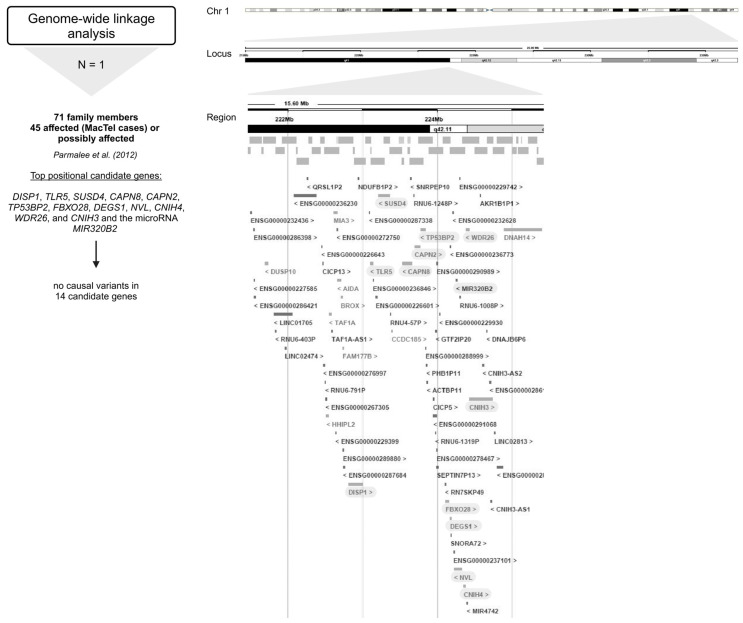
Overview of the genes encompassing the 16.5 Mb region at 1q41–42 investigated in genome-wide linkage analysis [24]. Below the approach, the number of studies (N) is given, followed by the number of study participants (MacTel cases and controls) (text in bold), the publication, and the candidate genes analyzed (marked gray) [24]. Created in BioRender.

**Figure 5 ijms-26-00684-f005:**
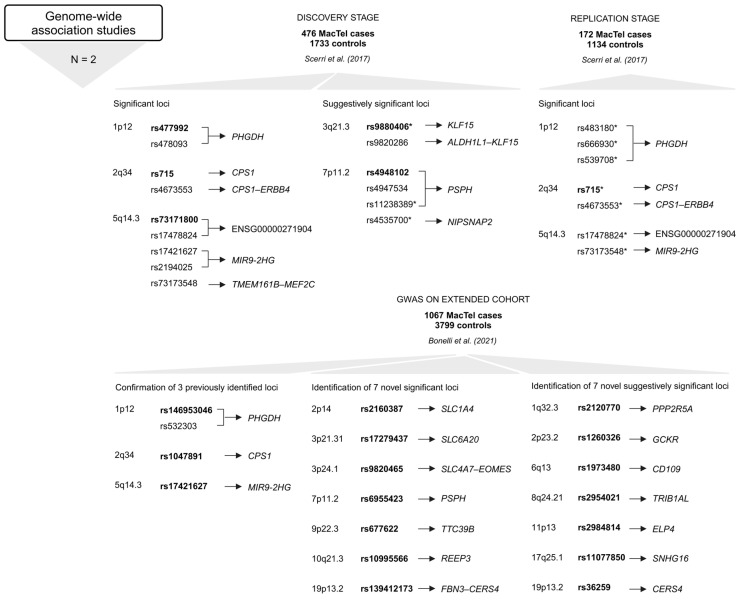
Variants and the most GW significant variants (rsIDs in bold) at each locus in genes or intergenic regions identified in GWAS [27,28]. Below the approach, the number of studies (N) is given, followed by the number of study participants (MacTel cases and controls) (text in bold) and the publication. The upper part refers to the first GWAS [28], and the lower part relates to the second GWAS [27]. * Variants that were genotyped in the replication stage [28]. Created in BioRender.

**Figure 6 ijms-26-00684-f006:**
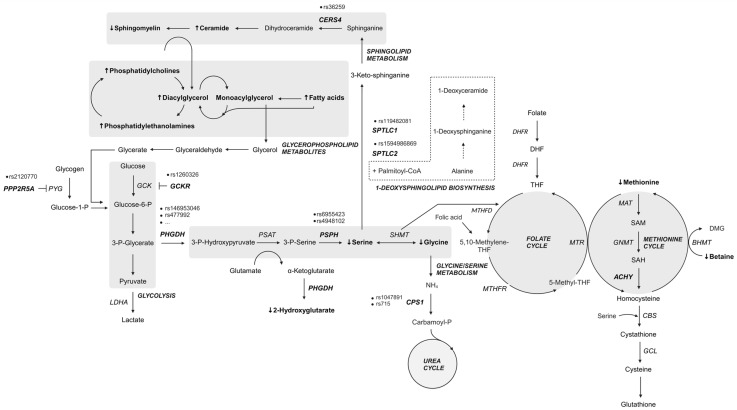
Summary of metabolic pathway genes and metabolites altered in patients with MacTel. Genes with variants most commonly reported in MacTel patients (bold, italic); elevated serum metabolites [29] (bold, ↑); reduced serum metabolites [29] (bold, ↓); dashed border: alternative pathway. Created in BioRender.

**Table 1 ijms-26-00684-t001:** Development of MacTel classification [2,17,18].

Gass and Blodi (1993) [2]	Yannuzzi et al. (2006) [17]	Chew et al. (2023) [18]
Group 1 1A: Visible and exudative IJRT 1B: Visible, exudative, and focal IJRT	Type 1 (aneurysmal)	
Group 2 2A: Occult and non-exudative IJRT Stage 1: Diffuse hyperfluorescence Stage 2: Reduced parafoveolar retinal transparency Stage 3: Dilated right-angled venules Stage 4: Intraretinal pigment clumping Stage 5: Vascular membranes	Type 2 (perifoveal) Non-proliferative perifoveal telangiectasia Proliferative perifoveal telangiectasia	Grade 0: No EZ break/No pigmentation/No OCT HR Grade 1: Non-central EZ break/No pigment/No OCT HR Grade 2: Central EZ break/No pigment/No OCT HR Grade 3: Non-central pigment/No, non-central, or central EZ/No OCT HR Grade 4: OCT HR/EZ break (either central or non-central)/No pigment Grade 5: Central pigment/No exudative neovascularization/EZ present or not gradable Grade 6: Neovascularization (exudative) ± central pigment
2B: Juvenile occult familial IJRT		
Group 3 3A: Occlusive IJRT 3B: Occlusive IJRT with CNS vasculopathy		

Classification of MacTel is indicated with gray background. Abbreviations: idiopathic juxtafoveolar retinal telangiectasia (IJRT); ellipsoid zone (EZ); optical coherence tomography (OCT); hyper-reflectivity (HR); central nervous system (CNS).

**Table 2 ijms-26-00684-t002:** Clinical reports of MacTel in families and associated primary conditions [2,6,11,36,37,38,39,40,41,42].

Affected Individuals (Age (Years))	Associated Primary Condition	Reference
Sisters (46 ^1^ and 56 ^2^)	^1^ Blurring of vision; ^2^ Slightly distorted vision after a car accident	[36]
Brothers (65 ^1^ and NS)	^1^ T2DM, mild non-proliferative DR, systemic hypertension, asteroid hyalosis, pigment epithelial hyperplasia, and mild macular edema	[6]
3/92 patients were siblings The disease is present in 2/89 families	14/92 patients had hypertension; other accompanying diseases were polycythemia vera, coronary artery disease, and borderline diabetes in one case each; two had coronary artery disease, and one had renal insufficiency associated with Alport’s disease	[2]
Sisters (49 ^1^ and 48 ^2^)	^1^ Reduced vision; ^2^ Blurring of distance vision	[37]
Daughter (29 ^1^) of the affected father (58 ^2^)	^1^ T2DM, decreased vision; ^2^ Macular edema	[38]
*** Monozygotic twins, sisters (64 ^1,2^)**	^1,2^ Vision loss	[40]
*** Monozygotic twins, sisters (68 ^1/2^)**	^1^ Bilateral metamorphopsia; ^2^ T2DM, amblyopia	[41]
*** Monozygotic twins, sisters (63 ^1/2^)**	^1^ Metamorphopsia, subretinal NV; ^2^ Metamorphopsia	[42]
Daughters (41 and 45) of affected mother (68 ^1^) Brother (61) of the affected sister (74 ^2^) *** Monozygotic twins, sisters (56 ^3^)** *** Monozygotic twins, brothers (56 ^4/5^)**	^1^ Blurring of vision; ^2^ Blurring of vision, T2DM; ^3^ Bilateral blurred vision, phototherapeutic keratectomy; ^4^ Decrease in vision, developed T2DM; ^5^ T2DM	[11]
Son (45 ^1^) of the affected mother (79 ^2^) Son (62 ^3^) of the affected mother (76 ^4^) Sister (65 ^5^) and brother (78 ^6^)	^1^ Reading difficulties, metamorphopsia, T2DM, mild DR, underwent coronary bypass at age of 40; ^2^ Loss of vision, maculopathy of unknown origin, T2DM, DR; ^3^ Visual loss, obesity, arterial hypertension, phlebothrombotic event at age of 57, hyperhomocysteinemia, antiphospholipid syndrome, T2DM; ^4^ T2DM, reading difficulties, maculopathy in the absence of DR; ^5^ Metamorphopsia, T2DM; ^6^ T2DM	[39]

Superscript numbers indicate the associated primary condition in the affected individual (family member). * Twin studies (text in bold). Abbreviations: diabetes mellitus type 2 (T2DM); diabetic retinopathy (DR); neovascularization (NV); not specified (NS).

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
