# Peer review of "Genetic Background of Macular Telangiectasia Type 2"

_ijms, 2025, doi:10.3390/ijms26020684_

Round 1
Reviewer 1 Report
Comments and Suggestions for Authors
I truly appreciate the authors for writing the review on "Genetic background of macular telangiectasia type 2" and I am glad to review this article. However, I would like to suggest the authors to work on this more and rewrite some sections to make it more understandable to the readers. Though the authors have tried to add lots of information, but it seems very unorganized and special attention needs to be given on the figures (1-3) to make it clearer. My comments are below:
Line 15: In aim to raise the disease.... please check the error
Line 69-71: This paragraph is very contradicting in terms of treatment and looks repeated.
Line 41: Reference missing
Table 1: This needs to be redrawn
Line 76: The prevalence of MacTel is estimated to be 0.0045–0.1%.... if this can be given in number like 1 in 1000 something like this.
Line 94: Reference missing
Line 96: Online libraries.... or online database
Line 99-101: It will be good to show the genetic variants on the genome or coding sequence to make it figurative.
Table 2: There is no nothing in several family members and forward slash (/) put. Also, Brothers (651 and /), I couldn't get this. (681/2)..why 1/2 ?
Line 155: variants in CFH and CFB genes.... provide full form and also can mention part of complement pathway. Same for the ATM gene also.
It will be good to provide full form for the abbreviated candidate genes. Lots of place's abbreviations are used without giving the full name. The authors must check this throughout the manuscript.
Line 355: the researchers found that an increase of 1 standard deviation in...I coudn't get this
Line 363: Overall, these and other studies indicated...check this.
Line 398-401: Reference missing
Figure 5 is important part of the paper but seems very blurred. Redraw this. I would recommend working on all the figures to make it clearer and more readable.
Comments on the Quality of English LanguageThe quality of the English needs to be improved, grammatical errors need to be checked, and proper proofreading needs to be done.
Author Response
Comment 1: I truly appreciate the authors for writing the review on "Genetic background of macular telangiectasia type 2" and I am glad to review this article. However, I would like to suggest the authors to work on this more and rewrite some sections to make it more understandable to the readers. Though the authors have tried to add lots of information, but it seems very unorganized and special attention needs to be given on the figures (1-3) to make it clearer.
Response 1: We thank the reviewer for the comments, which have helped to improve the manuscript. Figures have been reorganised and additionally described to make them clearer. A circular plot visualizing the genomic variants found in MacTel patients (summarized in Table S1–S4) on the genome has been added as Figure 2 in the revised manuscript. We provide point-by-point responses to the reviewer’s comments bellow. All corrections are indicated in the manuscript with track-changes. The manuscript has been proofread, the corrections are indicated in the revised manuscript with track-changes.
My comments are below:
Comment 2: Line 15: In aim to raise the disease.... please check the error
Response 2: Corrected to: »To better understand the molecular milieu of the disease,…«
Comment 3: Line 69-71: This paragraph is very contradicting in terms of treatment and looks repeated.
Response 3: This section was corrected as follows: »While intravitreal treatment with vascular endothelial growth factor inhibitors is available to control exudative neovascularization [19], there are currently no therapies for the neuro-degenerative changes associated with the disease.«
Comment 4: Line 41: Reference missing
Response 4: Reference added - [3]; Charbel Issa et al., 2013.
Comment 5: Table 1: This needs to be redrawn
Response 5: Table 1 was reformatted.
Comment 6: Line 76: The prevalence of MacTel is estimated to be 0.0045–0.1%.... if this can be given in number like 1 in 1000 something like this.
Response 6: Corrected as follows: »The prevalence of MacTel is estimated at 0.0045–0.1% (4.5–100 cases per 100,000 individuals) [20, 21].«
Comment 7: Line 94: Reference missing
Response 7: Reference added - [29]; Bonelli et al., 2020.
Comment 8: Line 96: Online libraries.... or online database
Response 8: Corrected as follows: »The online databases PubMed and Google Scholar were searched for…«
Comment 9: Line 99-101: It will be good to show the genetic variants on the genome or coding sequence to make it figurative.
Response 9: A circular plot visualizing the genomic variants found in MacTel patients (summarized in Table S1–S4) on the genome has been added as Figure 2 in the revised manuscript.
Comment 10: Table 2: There is no nothing in several family members and forward slash (/) put. Also, Brothers (651 and /), I couldn't get this. (681/2)..why 1/2 ?
Response 10: Modified in the manuscript with track changes - the reference in French (Putteman et al., 1984) was removed. In the case of monozygotic twins (the same age) the associated primary condition can be either of the first twin or second twin (therefore »1/2« - one or another twin brother/sister), e.g. in the reference 39 both twins were reported to have a 1,2vision loss - therefore: »1,2«; while in the reference 40, the first twin sister had 1bilateral metamophopsia and a second twin sister had 2T2DM and amblyopia - therefore: »1/2«. The twin studies for MacTel are now highlighted in Table 2 in bold.
Comment 11: Line 155: variants in CFH and CFB genes.... provide full form and also can mention part of complement pathway. Same for the ATM gene also.
Response 11: This section has been updated as follows: »Barbazetto et al. (2008) investigated variants in ATM serine/threonine kinase (ATM), age-related macular degeneration (AMD)-associated variants in complement factor H (CFH), complement factor B (CFB) and 10q26 loci in a cohort of 30 MacTel patients [23]. The ATM gene is crucial for DNA damage response and repair (genome stability), while CFH and CFB are involved in the alternative pathway of complement activation and play a role in the immune response [42, 43].«
Comment: 12: It will be good to provide full form for the abbreviated candidate genes. Lots of place's abbreviations are used without giving the full name. The authors must check this throughout the manuscript.
Response 12: Full names of candidate genes are provided in the supplementary tables (Table S1–S4).
Comment 13:Line 355: the researchers found that an increase of 1 standard deviation in...I coudn't get this
Response 13: Corrected as follows: »…, the researchers found that higher serine levels reduced the likelihood of MacTel development by 95%,…«
Comment 14: Line 363: Overall, these and other studies indicated...check this.
Response 14: Corrected as follows: »Overall, the studies indicated…«
Comment 15: Line 398-401: Reference missing
Response 15: Reference added - [3, 11]; Charbel Issa et al., 2013; Gillies et al., 2009.
Comment 16: Figure 5 is important part of the paper but seems very blurred. Redraw this. I would recommend working on all the figures to make it clearer and more readable.
Response 16: Figures with 600 dpi were uploaded together with the manuscript and are available as separate files.
Reviewer 2 Report
Comments and Suggestions for Authors
Thank you for inviting me to review this paper. With the increasing focus on genetic diagnostic techniques, this study has objectively analyzed molecular and genetic findings of Macular Teleangiectasia type 2, which is still a mysterious disease. I find it very interesting.
Minor revisions suggested:
Abstract
Line 22: genetic alterations ( alternations)
1. Introduction:
Line 61: retinal telangiectasia (telangiectasis)
Line 92: it would be interesting to describe how these metabolic alterations may be linked
to the onset of MacTel. Please discuss further.
Line 96: Please describe literature research process ( eg. MeSH terms)
2. Key genes, genomic regions, and metabolic pathways associated with MacTel
Line 96: Please describe literature research process ( eg. MeSH terms)
Line 141: Please add references supporting this sentence
3. Conclusions:
Line 461- 462: are there any evidence that suggest L-serine supplementation could be
used to slow vision loss in MacTel patients?
Comments on the Quality of English LanguageQuality of english language could be improved.
Author Response
Comment 1: Thank you for inviting me to review this paper. With the increasing focus on genetic diagnostic techniques, this study has objectively analyzed molecular and genetic findings of Macular Teleangiectasia type 2, which is still a mysterious disease. I find it very interesting.
Response 1: We thank the reviewer for the comments, which have helped to improve the manuscript. We provide point-by-point responses to the reviewer’s comments bellow. All corrections are indicated in the manuscript with track-changes. The manuscript has been proofread, the corrections are indicated in the revised manuscript with track-changes.
Minor revisions suggested:
Abstract
Comment 2: Line 22: genetic alterations ( alternations)
Response 2: Corrected as follows: »Genetic alterations can disrupt…«
- Introduction:
Comment 3: Line 61: retinal telangiectasia (telangiectasis).
Response 3: Corrected all over the manuscript to: »telangiectasia« with track changes.
Comment 4: Line 92: it would be interesting to describe how these metabolic alterations may be linked to the onset of MacTel. Please discuss further.
Response 4: We described metabolic alternations in section 2.6 Implications of risk-associated metabolic pathways on MacTel pathogenesis as follows: »The retina is very metabolically active and the transport of serine across the blood-retinal barrier into the RPE or across the endothelial cells to the neurosensory retina is supposedly inadequate. The additional supply of serine to the photoreceptors and the inner retina is provided by the RPE and the Müller glial cells (MGCs). Both cells can syn-thesize serine. MGCs in the macula show increased PHGDH levels, glutathione and glycine production and are more susceptible to induced stress [65]. Many retinopathies are associated with the loss of MGCs, including MacTel, and the loss of MGCs leads to im-paired PHGDH activity and reduced serine levels. The supply of glycine and glutathione, and thus serine, is essential for reactive oxygen species mitigation system of the rods and cones. Consequently, reduced serine levels have negative impact on these repair mechanisms. Lower levels of D-serine, which functions as a neurotrophic factor and coagonist for N-methyl-D-aspartate receptors, probably promote neurodegenerative changes in MacTel patients [65]. Phosphatidylserine is important for the recognition and phagocytosis of correct por-tion of the outer segments. Inefficient or delayed phagocytosis of the photoreceptor outer segments can lead to abnormalities of the neurosensory retina and RPE. Severe serine defi-ciency and dysregulated lipid metabolism can lead to increased and leaky vasculature and contribute to vascular abnormalities [65].«
Comment 5: Line 96: Please describe literature research process ( eg. MeSH terms)
Response 5: The Introduction section in the manuscript was extended as follows: »The online databases PubMed and Google Scholar were searched for the relevant literature using keywords: “macular telangiectasia”, “macular telangiectasia type 2”, “MacTel”, plus “genetics”, “genomics” and “family members”. The titles and abstracts were screened first, followed by an eligibility assessment of the full text. Inclusion criteria: publications written in English referring to human studies were included. Exclusion criteria: Macular telangiectasia type 1 and 3; conference abstracts, and master's and doctoral theses were not included in the review.«
- Key genes, genomic regions, and metabolic pathways associated with MacTel
Comment 6: Line 96: Please describe literature research process ( eg. MeSH terms)
Response 6: Added in the manuscript (Introduction) as described above. Additional information about variants are presented in the supplementary tables (S1–S4 - gene symbol, gene name, NCBI ID, position, reference, HGVS.g, HGVS.c, HGVS.p, major and minor allele frequencies, clinical significance, variant type and lenght, and most severe consequence).
Comment 7: Line 141: Please add references supporting this sentence
Response 7: Reference added - [11, 24]; Gillies et al., 2009; Parmalee et al., 2012.
- Conclusions:
Comment 8: Line 461- 462: are there any evidence that suggest L-serine supplementation could be used to slow vision loss in MacTel patients?
Response 8: A phase 2a clinical study is ongoing: »Serine and Fenofibrate Study in Patients With MacTel Type 2 (SAFE)« (NCT04907084), and investigates the effect of serine supplementation and fenofibrate treatment on serum deoxysphingolipids in MacTel patients. No results have been published to date. The Conclusion section of the manuscript was extended as follows: »An ongoing phase 2a clinical study (NCT04907084) in MacTel patients is investigating the effect of serine supplementation and fenofibrate treatment, but up to our knowledge no results have yet been published.«
Reviewer 3 Report
Comments and Suggestions for Authors
The review is comprehensive, focusing on the genetic and metabolic intricacies of MacTel, addressing both research gaps and clinical implications. However, I have a few recommendations for improvement:
1. The authors could explicitly discuss the limitations of their review. This would provide a balanced perspective and highlight areas requiring further investigation.
2. Although this is not a systematic review, I suggest the authors describe their search methodology to ensure all relevant studies were included. Providing details about the searching strategy, inclusion criteria, and exclusion criteria for the studies reviewed would enhance the transparency and reliability of their findings.
Author Response
Comment 1: The review is comprehensive, focusing on the genetic and metabolic intricacies of MacTel, addressing both research gaps and clinical implications. However, I have a few recommendations for improvement:
Response 1: We thank the reviewer for the comments, which have helped to improve the manuscript. We provide point-by-point responses to the reviewer’s comments bellow. All corrections are indicated in the manuscript with track-changes.
Comment 2: 1. The authors could explicitly discuss the limitations of their review. This would provide a balanced perspective and highlight areas requiring further investigation.
Response 2: The Conclusion section in the manuscript was extended to discuss the limitations and highlight areas requiring further investigation: »In this review, we focus on the current knowledge about the genetic background of the disease. Several studies focus on the metabolomic alternations in affected patients, but not on the genetic background of these patients.«
»Further studies linking the metabolomic alternations to the genetic background of the patients will help to further clarify the genetic background of the disease.«
Comment 3: 2. Although this is not a systematic review, I suggest the authors describe their search methodology to ensure all relevant studies were included. Providing details about the searching strategy, inclusion criteria, and exclusion criteria for the studies reviewed would enhance the transparency and reliability of their findings.
Response 3: The Introduction section in the manuscript was extended as follows: »The online databases PubMed and Google Scholar were searched for the relevant literature using keywords: “macular telangiectasia”, “macular telangiectasia type 2”, “MacTel”, plus “genetics”, “genomics” and “family members”. The titles and abstracts were screened first, followed by an eligibility assessment of the full text. Inclusion criteria: publications written in English referring to human studies were included. Exclusion criteria: Macular telangiectasia type 1 and 3; conference abstracts, and master's and doctoral theses were not included in the review.«
Round 2
Reviewer 1 Report
Comments and Suggestions for Authors
Dear Authors,
I went through your revised manuscript, and I am happy with the revised version provided by you based on my comments. I have no comments further in this revised manuscript.